First photographic evidence of oceanic manta rays (Mobula birostris) at two locations in the Fiji islands

Gordon Luke luke.gordon@mantatrust.org 1
Vierus Tom 2
1 Manta Trust , Dorchester , United Kingdom
2 Suva , Fiji Islands
Ron Santiago
Electronic publication date: 2022 Sep 7
Publication date: 2022
Volume: 10
Electronic Location ID: e13883
Received 2022 Apr 22; Accepted 2022 Jul 21
Copyright: ©2022 Gordon and Vierus
Copyright year: 2022
Copyright holder: Gordon and Vierus
License: This is an open access article distributed under the terms of the Creative Commons Attribution License, which permits unrestricted use, distribution, reproduction and adaptation in any medium and for any purpose provided that it is properly attributed. For attribution, the original author(s), title, publication source (PeerJ) and either DOI or URL of the article must be cited.
License URL: https://creativecommons.org/licenses/by/4.0/

Keywords: Ecology, Citizen science, Mobulidae, Fiji islands, Manta ray, Elasmobranch

Funding: The authors received no funding for this work.

==============================
Until the revision of the genus Manta in 2009, when a second manta species (Manta alfredi) was resurrected based on morphological and meristic data, all available records in Fijian literature were recorded as Manta birostris. Subsequently, documented sightings were recorded as M. alfredi. Another reclassification of the genus Manta was undertaken in 2018 when both manta ray species (Manta alfredi, Manta birostris) were moved to Mobula based on phylogenetic analysis. Here, we present the first unequivocal evidence of oceanic manta ray (Mobula birostris) occurrence in Fijian waters. In November 2018, two individuals were sighted foraging in Laucala Bay, a large lagoon adjacent to Suva, the capital city of Fiji. Subsequently, three more individuals were sighted in December 2018, two individuals in July 2020, at least six individuals were observed in November 2021, and eight individuals in May/June 2022, all foraging in the same geographical area. Unique ventral identification patterns could be obtained for nine individuals, and all nine individuals have been re-sighted since first identification, with one individual being documented in 2018, 2020, 2021 and 2022. Two additional individuals were recorded in the Yasawa Island Group in the west of Fiji while passing through and foraging in a channel between Drawaqa and Naviti Island in April and September 2020. We provide photographic identification of ten M. birostris individuals from two sites and discuss our findings in the context of local environmental parameters and other recorded sightings in the South Pacific region. In light of the global extinction risk of M. birostris and the recent reclassification from Vulnerable to Endangered on the Red List of Threatened Species, the expansion of their known distribution range to Fijian waters and the recurrence of individuals over consecutive years in the same location adds valuable information for the development of effective and data-driven conservation strategies.

Introduction

Manta rays (Mobula spp.) are large and charismatic zooplanktivorous elasmobranchs found in tropical and subtropical waters throughout the world (Marshall et al., 2020; Marshall et al., 2019: Fig. 1). The two recognised species, Mobula birostris (oceanic manta ray) and Mobula alfredi (reef manta ray) belong to the family Mobulidae together with seven other ray species. Until 2009, the scientific consensus only included one manta ray species (Manta birostris). This changed after a review by Marshall, Compagno & Bennett (2009), when a second species, Manta alfredi, was resurrected based on morphological and meristic data. Nine years later, a phylogenetic study by White et al. (2018) sequenced mitochondrial, and nuclear DNA of the complete taxon, and based on the results proposed moving both manta ray species from the genus Manta to the genus Mobula, changing their nomenclature to Mobula alfredi and Mobula birostris. The authors noted that by solely sequencing mitochondrial DNA, both species were indistinguishable, but when incorporating nuclear DNA in combination with morphological data, results indeed supported the proposed taxonomic changes. Speciation has occurred relatively recently in evolutionary terms, and the close genetic relationship is likely a result of post-divergence gene flow through hybridisation (Kashiwagi et al., 2012). Interestingly, a recent study by Hosegood et al. (2020) additionally presents evidence of a putative third manta ray species in the Gulf of Mexico, indicating potential further taxonomic changes to the Mobula genus.

Figure 1 Map showing the global distribution ranges of M. birostris (top) and M. alfredi (bottom; adapted from IUCN, 2022).

Source: IUCN (2022), ESRI, GEBCO, DeLorme, NaturalVue.

The reef manta ray, M. alfredi, is generally observed in nearshore areas or in the vicinity of continental coastlines, exhibiting small home ranges and a high degree of site fidelity (Couturier et al., 2011), albeit exceptions have been observed, such as a reef manta ray recorded at Cocos Island nearly 6,000 km from the nearest confirmed sighting (Arauz et al., 2019). The more widely distributed oceanic manta ray, M. birostris, occurs in all three major oceans (Marshall et al., 2020), often observed in pelagic environments, such as offshore seamounts, pinnacles or oceanic islands (Marshall, Compagno & Bennett, 2009; Fig. 1). Similar to other elasmobranchs, targeted and untargeted fisheries coupled with life-history traits, such as slow growth, late maturation, long gestation periods and low fecundity render both manta species particularly vulnerable to overexploitation (Couturier et al., 2012; Dulvy et al., 2014; Pardo et al., 2016). Declining populations due to the aforementioned factors led to conservation concerns for both species, with M. alfredi listed as Vulnerable and M. birostris listed as Endangered to Extinction on the IUCN Red List of Threatened Species (Marshall et al., 2019; Marshall et al., 2020). On a national level, both manta ray species are legally protected in Fiji by the ‘Endangered and Protected Species Act (EPS)’ adopted in 2002, which requires permits to trade or land species listed in Appendix I, II or III of CITES, the ‘Convention on International Trade on Endangered Species of Wild Fauna and Flora’ (Fiji Government, 2002). Similarly, Fiji’s ‘Offshore Fisheries Management Act (OFMA)’ adopted in 2012, forbids the killing, taking, landing, selling and transporting of species listed in Appendix I and II of CITES (Fiji Government, 2012). Besides introducing national legislation, Fiji has repeatedly advocated for more protection on an international level. For example, in 2014, Fiji led the successful proposal for inclusion of all Mobula species in Appendix II of the ‘Convention on the Conservation of Migratory Species’ (Convention of Migratory Species, 2014) and in 2016, the successful proposal for inclusion of the same group in Appendix II of CITES (Convention on International Trade in Endangered Species of Wild Fauna and Flora, 2016). Fiji reaffirmed their domestic ambitions at the UN Ocean Conference in New York in 2017 by committing to the “conservation and management of all species of sharks and rays and their critical habitats within Fijian waters” (United Nations, 2017). Surprisingly, to date, there are no documented records of M. birostris in Fiji’s waters after the resurrection of M. alfredi in 2009 besides brief mentions in the catch statistics of Fijian longline pelagic fisheries (Piovano & Gilman, 2017). Earlier records of Manta birostris can be found in Fijian literature, for example, a 300–400 kg specimen recorded off Rotuma in 1983 (Fijian Fisheries Division, 1983). While some reliable reef manta aggregation sites are known throughout the country and opportunistic Mobula spp. sightings are commonly reported by recreational divers, fishermen and tourism operators, detailed information on habitat preferences and distribution within the country is generally lacking. Several Fiji-based tourism operators offer reef manta ray snorkelling activities, most notably Barefoot Manta, an ecotourism resort based on Drawaqa island in the Yasawa Island Group, approximately 40 km north-west from Viti Levu. The island group consists of 11 main volcanic islands running 90 km to the north-east (Ward & Beggs, 2007). A 250 m long, 300 m wide and approx. A 7 m deep channel located towards the southern end of Drawaqa Island and the largest island in the chain, Naviti Island, is a known reef manta ray aggregation site. From May to October aggregations of up to 15 reef manta rays can be observed in the channel (Murphy, Campbell & Drew, 2018). In addition to feeding on plankton, the reef manta rays also opportunistically utilise a cleaning station in the passage. Similarly, the waters off Kokomo Private Island Fiji, a luxury resort based in the south of the country on Yaukuve Levu Island, part of an island chain to the north of Kadavu Island, are home to several foraging sites and cleaning stations with regular sightings from April-December and a peak in sightings from May–October. Large aggregations have been recorded at these sites with 65+ individuals foraging at the same time, currently the largest aggregation of reef manta rays known in Fiji (L Gordon, 2020, pers. obs). Manta Project Fiji (MPF), established in 2012 as an affiliate of the Manta Trust, has been cataloging reported manta ray sightings across the country and currently manages a database containing 425 identified M. alfredi individuals. Prior to this study, no oceanic manta rays had been reliably confirmed through photographs or video in Fijian waters and were absent from the database.

Adjacent to Suva, Fiji’s capital, lies Laucala Bay, a relatively flat coastal lagoon enclosed by a barrier reef (Fig. 2). Here, reef manta rays (Mobula alfredi) are commonly observed foraging and at least one individual was captured here by local fishermen in August 2021. Laucala Bay lies between the Suva peninsula in the west (where a hilly environment separates it from Suva Harbour) and the delta of Fiji’s largest river, the Rewa, in the east (Fig. 2). The tidal range of the bay lies between 0.9–1.33 m, with an average depth of 9–15 m and a maximum depth of 30–40 m (Morrison, Narayan & Gangaiya, 2001; Koliyavu et al., 2021). During high tide, Laucala Bay’s surface area extends to 4,500 ha, with several emerging mudflats and sandbanks shrinking it to 3,900 ha during low tide (Morrison, Narayan & Gangaiya, 2001). Besides being located adjacent to the Rewa delta, several rivers feed into the bay area shedding large amounts of freshwater into the area with limited exchange towards the oceans due to the reef system sheltering it from the open ocean (Koliyavu et al., 2021). Additionally, the bay receives treated domestic, commercial and industrial wastewater discharged from the Kinoya sewage treatment plant into the northern part of the bay (Fig. 2; Ferreira et al., 2020). This paper discusses all recorded M. birostris sightings in Fijian waters to date, presenting photographic evidence of nine M. birostris individuals foraging in Laucala Bay near Suva and two additional M. birostris sightings in the Yasawa Island Group and explores the sightings in relation to local environmental parameters. It thus provides the first unequivocal evidence of oceanic manta ray occurrence within Fijian waters.

Figure 2 Map of the greater Suva area which includes Laucala Bay, Suva city, Suva Harbour, the Rewa River and the Rewa Estuary.

The foraging area of the observed oceanic manta rays is located at the southern-western end of Laucala Bay (highlighted) near one of the channels in the barrier reef.

Material & Methods

Based on frequently reported sightings of rays by local citizens over the years in Laucala Bay off Suva, Fiji’s capital, the main author of this study and project manager of MPF started opportunistic surveys in 2018. The surveys were focused on the reported occurrence area and continued when possible throughout the next four years mostly within November, December and July, the months with the highest ray sighting reports. At the end of November 2021, after several recreational boaters had sent videos of rays in Laucala Bay, targeted surveys were undertaken on eight consecutive days and subsequently sporadically continuing until June 2022 when possible. Utilising a fibreglass boat, the Laucala Bay area was systematically explored by slowly cruising parallel to the coast and scanning the horizon for signs of manta ray activity. Surveys were timed to coincide with the arrival of high tide, as manta ray activity and sightings seem to be at their peak and the first 45–60 min thereafter. However, manta ray activity was also observed irrespective of tidal activity. Re-sightings are defined as sightings of the same individual on different calendar days while sightings of the same individual on the same day are recorded as one sighting record. A drone was utilised to monitor a larger area and to attempt to take aerial photographs and/or videos of their ventral side while feeding below the surface. Sighting coordinates were recorded on a boat GPS when a manta ray was in close proximity to the boat (<30 m) or extracted from the drone metadata when flying directly above the manta ray. Besides the opportunistic surveys in the Laucala Bay area, daily manta ray surveys were also undertaken from April to October in the channel between Drawaqa and Naviti Island within the Yasawa Island Group (17.16335°S 177.19270°E; Fig. 3) to coincide with high tide when reef manta ray foraging activity peaks at this site (L Gordon, 2021, personal observation). Collected drone and underwater photographs and video frame grabs were colour and contrast-enhanced utilising Adobe Lightroom and subsequently analysed for unique M. birostris identification marks using the key provided in Marshall, Compagno & Bennett (2009): coloration of the dorsal shoulder patches and pectoral fins (1), chevron-shaped marking anterior to the dorsal fin (2), dark spots anterior to the 5th gill slit (3), dark grey coloration of ventral pectoral fin margins (4) and dark coloration (grey to black) of the ventral mouth region (5). Individuals were then added to Manta Project Fiji’s database, which currently encompassed more than 425 unique identifications of M. alfredi and eleven identifications of M. birostris, ten of which are presented in this paper.

Figure 3 Map of the Yasawa Islands and Viti Levu, Fiji’s main island. Inset (top right) displays a close-up map of the Manta Channel between Drawaqa Island and Naviti Island.

Results and Discussion

During opportunistic sampling of Laucala Bay spanning from December 2018 to June 2022, at least 11 M. birostris individuals were observed, with nine individuals being photographically identified (Fig. 4). The two additional individuals were able to be differentiated by dorsal markings and injuries (such as a shark bite), however, no ventral identifications could be collected. Notably, all nine identified individuals have subsequently been re-sighted at the same site, with FJ-MB-0001 having been sighted nine times since December 2018 (Fig. 5). All specimens presented in this study displayed repeated somersault and surface feeding before leaving the area approx. 45–60 min after high tide. In addition to the Laucala Bay sightings, in April and September 2020, two M. birostris individuals were filmed by Mathjis Carmen in the channel between Drawaqa and Naviti Island in Fiji’s Yasawa Island Group, a known feeding and cleaning site for reef manta rays M. alfredi (Fig. 6). Only the individual recorded in September 2020 was identified while foraging in the channel and was re-sighted foraging at the same location the next day. Notably, this was the first time observing M. birostris in this area despite daily sampling between April and October for the past nine years, suggesting this location was visited opportunistically and does not represent a reliable observation site for M. birostris. Contrastingly, repeated sightings in Laucala Bay over at least four years indicate a reliable observation area. While ray activity in the bay reported by recreational users or fishermen may be attributed to visually similar M. alfredi individuals, either scenario provides interesting insights, as shared foraging grounds between M. birostris and M. alfredi add to the knowledge of existing locations where both species occur in microsympatry (co-occurrence at the same site; Kashiwagi et al., 2011). During the course of fieldwork, multiple observations of microsympatry were recorded, with both M. alfredi and M. birostris feeding in very close proximities (>20 m; Fig. 7). No direct interactions between the two species were observed, apart from one instance when almost swimming into each other, which startled both animals.

Figure 4 Identification photographs of nine M. birostris individuals sighted in Laucala Bay adjacent to Suva, Fiji’s capital city.

With comparison to M. alfredi. Manta identification names are shown at the bottom left, e.g., ‘FJ-MB-0001’. Two individuals (FJ-MB-0001, FJ-MB-0002) were identified underwater while the remaining seven were photographed or filmed utilising a drone. White arrows (A) and (C) indicate key morphological features for M. birostris: (A) shows the distinctive grey V-shaped margin along the posterior edge of the pectoral fins; and (C) shows the white dorsal shoulder markings that form two mirror image right-angled triangles. Ventral spots clustered around the lower abdomen region which are used for identification are indicated by (B). White arrows (D) and (E) indicate key morphological features for M. alfredi: (D) shows ventral identification spots clustered between the gill slits and across trailing edge of pectoral fins; and (E) shows the white blurred ‘V’ dorsal markings.FJ-MB-0001 image shows two sightings, the original sighting (inset, bottom right) from 02.12.2018 and the most recent from 24.11.2021. Photographs taken by Tom Vierus, Luke Gordon and Cliona O’Flaherty.

Figure 5 Visualisation of all re-sightings of the nine identified M. birostris individuals in Laucala Bay between December 2018 and June 2022.

Figure 6 M. birostris individuals sighted in the channel between Drawaqa and Naviti Island in the Yasawa Group in north-western Fiji.

Only one individual could be filmed from below revealing its unique identification pattern (FJ-MB-0003). White arrows (A) and (C) indicate key morphological features for M. birostris: (A) shows the distinctive grey V-shaped margin along the posterior edge of the pectoral fins; and (C) shows the white dorsal shoulder markings that form two mirror image right-angled triangles. Ventral spots clustered around the lower abdomen region which are used for identification are indicated by (B).

Figure 7 Images illustrating M. birostris and M. alfredi microsympatry in Laucala Bay. White arrows (A) and (B) indicate distinctive dorsal markings for: (A) M. alfredi; and (B) M.birostris.

A recent study focusing on nutrient measurements in Laucala Bay reported high chlorophyll-a concentrations (phytoplankton biomass), especially in the immediate coastal zones (Koliyavu et al., 2021). The authors of the study attribute the high values to the accumulation of nutrients from high riverine discharges and anthropogenic inputs, such as the effluents discharged from the Kinoya wastewater treatment plant in the north of the bay coupled with a low water outflow due to the barrier reefs restricting water exchange to and from the open ocean (Fig. 2). Notably, study sites within the inner bay zone that displayed the highest mean chlorophyll-a measurements are consistent with the observed foraging areas of M. birostris, suggesting the individuals are specifically targeting these areas to maximise their foraging success. While more surveys are needed to confirm the presence of M. birostris consistently over a longer time frame than the four years presented here, the current observational data and the spatio-temporal overlap of chlorophyll-a concentrations with manta occurrences suggests that Laucala Bay might be visited annually in at least November, December, June and July, presumably for feeding on zooplankton blooms following high phytoplankton concentrations (Koliyavu et al., 2021).

Despite the limited size of our dataset, the current re-sighting rate of 100% is exceptionally high for M. birostris. This is in stark contrast to other M. birostris populations, such as in Isla de la Plata, Ecuador (the largest population in the world) where the re-sighting percentage over the last ten years is very low (M Guerrero, 2022, personal communication). This difference in re-sighting rate and degree of site fidelity could possibly be attributed to geographical differences in habitat use and explained by a hypothesis surrounding metapopulation theory. The theory states that a larger population could be made-up of smaller sub-population units that interact through gene flow and migration, i.e., M. birostris individuals in Laucala Bay may be mostly resident (explaining the high re-sighting rate) with some individuals passing through on longer migrations. Recent studies investigating the genetic population structure in M. alfredi are presenting contrasting results; for example, in New Caledonia, M. alfredi were found to exhibit a fine-scale genetic population structure (Lassauce et al., 2022), but not in Mozambique (Venables et al., 2021), indicating the need for conservation assessments on a case by case basis. This may also translate to M. birostris, as evidenced in research by Stewart et al. (2016), which found significant population structure among M. birostris, and added that long-distance migrations are likely rare. On the other hand, Hosegood et al. (2020) failed to find any significant population structure, suggesting homogeneity within the global M. birostris population. Furthermore, unpublished tagging data (M Erdmann, 2022, personal communication) from Conservation International Aroterea, Manta Watch New Zealand and the New Zealand Department of Conservation infer that a proportion of the South Pacific population of M. birostris undertakes seasonal migrations between Fiji/Tonga and New Zealand (and possibly further afield). It remains to be seen whether migratory individuals traveling from New Zealand visit Laucala Bay or whether the individuals already sighted in Laucala Bay have made longer distance migrations between sighting dates or in years prior to sightings, potentially mixing as a larger population. In light of Fiji’s commitment to protect and manage critical habitats for rays and sharks at the UN Ocean Conference in New York in 2017 (United Nations, 2017) coupled with the extinction threat of oceanic manta rays, our findings provide valuable information to develop and advance protective measures to safeguard this species within Fijian waters. While the national ‘Endangered and Protected Species Act’ and the ‘Offshore Fisheries Management Act’ provide a legal framework for the protection of both manta species, the logistical difficulty of monitoring and enforcement remains to be solved. The occurrence of M. birostris in such close proximity to Fiji’s capital city Suva (estimated population of 256,000 including the Greater Suva area; Pratap, Mani & Prasad, 2020) makes this discovery especially noteworthy as increasing urban development will inevitably cause increasing pollution and boat traffic. Both factors have been shown to pose risks to foraging mantas (Couturier et al., 2012; Marshall et al., 2020). Furthermore, the coastal environments around the Suva peninsula constitute important habitats for a number of other threatened elasmobranch species besides M. alfredi and M. birostris. For example, the Rewa delta and river located to the immediate east of the city have been documented to constitute important pupping habitats for the Critically Endangered scalloped hammerhead shark Sphyrna lewini and the Vulnerable bull shark Carcharhinus leucas (Brown et al., 2016; Glaus et al., 2019). The close proximity to a fast-growing city such as Suva will therefore require a careful management approach to maintain the integrity of the surrounding coastal ecosystems.

Besides confirming M. birostris occurrence in Fijian waters for the first time, our findings suggest that Laucala Bay may represent a critical foraging habitat for the species rendering it an area of interest not only for Fiji but for the wider South Pacific region. In light of the critical conservation status of both manta ray species and their sympatric occurrence in Laucala Bay as well as in the Yasawa Island Group, we recommend further monitoring to build on our observations. We also suggest that increased local awareness of these findings could be helpful in obtaining additional data on ray sightings from recreational users of above-mentioned areas. Future research should incorporate visual sampling over the entire year and over multiple years to elucidate the temporal distribution of this species in the region and determine whether the visiting mantas represent a local population, individuals on longer journey migrations, or a mix of both. Furthermore, research methodologies that incorporate fine-scale and broad-scale movement tracking with genetic analysis would be helpful to further understand the population dynamics of manta rays that visit Laucala Bay.

Conclusions

This study provides the first unequivocal photographic evidence of M. birostris occurrence at two locations within Fiji’s Exclusive Economic Zone (nine individuals recorded in Laucala Bay and two individuals recorded near Drawaqa Island) with one of the observed mantas visiting over at least a four-year period and being sighted nine times (2018, 2020, 2021, 2022).

Supplemental Information

Supplemental Information 1 Raw sighting data of oceanic manta rays in Fiji

Each row represents a sighting record of an individual manta ray.

Click here for additional data file.

Supplemental Information 2 Ventral identification image showing unique ventral spots of FJ-MB-0001 taken from a drone in Nov 2021, Laucala Bay, Suva, Fiji Islands

Click here for additional data file.

Supplemental Information 3 Ventral identification image showing unique ventral spots of FJ-MB-0001 taken underwater in Dec 2018, Laucala Bay, Suva, Fiji Islands

Click here for additional data file.

Supplemental Information 4 FJ-MB-0002 raw image

Click here for additional data file.

Supplemental Information 5 FJ-MB-0003 raw image

Click here for additional data file.

Supplemental Information 6 FJ-MB-0004 raw image

Click here for additional data file.

Supplemental Information 7 FJ-MB-0005 raw image

Click here for additional data file.

Supplemental Information 8 FJ-MB-0006 raw image

Click here for additional data file.

Supplemental Information 9 FJ-MB-0007 raw image

Click here for additional data file.

Supplemental Information 10 FJ-MB-0008 raw image

Click here for additional data file.

Supplemental Information 11 FJ-MB-0009 raw image

Click here for additional data file.

Supplemental Information 12 FJ-MB-0010 raw image

Click here for additional data file.

Supplemental Information 13 Dorsal View (M. birostris)

Click here for additional data file.

Supplemental Information 14 Dorsal View (M. Alfredi)

Click here for additional data file.

Supplemental Information 15 Ventral View (M. alfredi)

Click here for additional data file.

We are grateful to the Fijian Government and the support from the Ministry of Education, Heritage and Arts and Ministry of Fisheries. We would also like to thank Cliona O’Flaherty, Mathjis Carmen and Barefoot Manta Resort for their contributions to the Manta Project Fiji database. The authors would also like to thank Marine Ecology Consulting, Daniel Lund and Timothy Stats for the logistical support to Manta Project Fiji.

Additional Information and Declarations

Competing Interests

Author Contributions

Data Availability

The authors declare there are no competing interests.

Luke Gordon conceived and designed the experiments, performed the experiments, analyzed the data, prepared figures and/or tables, authored or reviewed drafts of the article, and approved the final draft.

Tom Vierus performed the experiments, authored or reviewed drafts of the article, and approved the final draft.

The following information was supplied regarding data availability:

Raw data, including RAW sighting data and all RAW identification images, are available at Github:

https://github.com/mantaluke/birostris-laucala/issues.

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
