# Peer review of "First photographic evidence of oceanic manta rays (Mobula birostris) at two locations in the Fiji islands"

_PeerJ, doi:10.7717/peerj.13883_

## Round 0.1 · original submission · Major Revisions

I have read your manuscript and agree with Reviewer 1 in that it is an important contribution to the knowledge of Mobula birostris. Reviewer 1 made insightful comments that should help to improve the next version of the manuscript. Unfortunately, Reviewer 2 made a short review without providing any advice on how to improve the article. I also read the manuscript myself and my main comments are listed below. Minor additional comments are provided in the attached PDF.
Based on my assessment of the article, I conclude that major changes are still needed before it is accepted for publication.
Reviewer 1 was very thorough, so I ask you to read those comments carefully and address them, one by one, in the manuscript and in the response letter. I don't disagree with any of the comments and I believe these will only improve the manuscript.

My comments:
1. The Introduction and the Abstract need to mention the fact that prior to the resurrection of M. alfredi in 2009, all records of the genus Mobula from Fiji were reported as “M. birostris”. This information is crucial to correctly interpret the aim and scope of the study. Please, provide general information on the taxonomic history of both species.
2. The introduction should mention that one study have shown gene flow between M. alfredi and M. birostris (Kashiwagi et al., 2012) and another study (White et al., 2018) suggests that both species may actually represent a single one (i.e., M. alfredi is a junior synonym of M. birostris). The authors should comment on those studies, especially White et al. (2018) since the conspecificity of both species would change the implications of the new reported sightings.
3. Introduction. Considering the scope of the manuscript, the Introduction should be shortened. For example, lines 92–99 have information that is not necessary to understand and interpret the results. That section should be deleted.
4. Methodology: please, describe how were recorded the coordinates for the new sightings.
5. Methodology: for clarity, please add photographs of M. alfredi to Figure 3 or 4, pointing out the differences with M. birostris. The arrows in Figure 3 are quite helpful to show the characters used for species identification. It will be even better if the alternative states are similarly shown in M. alfredi.
6. Conclusions. Please delete sentences two and three, they are not conclusions.
7. Raw data. To adhere to the Journal policies, please upload to a public repository the raw photographic files on which the new records are based.

Reviewer 1 ·

Basic reporting

I was pleased to review this manuscript which is a step forward in increasing the knowledge regarding a fantastic but unfortunately endangered species. I found that this paper is straightforward, clear and well-written. The background supports the purpose of the research and the references are relevant and up to date. Figures and supplementary data look appropriate for the study. I think this study is beneficial not only for helping to fill the gap about the distribution range of the species by validating historical records with photographic evidence but for discussing about its ecology – information that is also limited.

Experimental design

This paper fits within the scope of the journal. I don’t have any major issues with the paper. All of my comments are small (mentioned below) and I think it should therefore be accepted after mostly minor revisions. For other comments please check the attached manuscript.

Lines 40-43. Besides talking about habitat preferences, I would like to see the distribution range of both species in more detail as this is relevant for the paper context. In my opinion, a map showing both distributions within the region could be a useful addition to the paper.

It is known that the oceanic manta ray is a migratory species. However, for some populations in Mexico and the Indo-Pacific there is evidence of fidelity and restricted migration rate for the species. Could you discuss a bit more about how your research can provide information to fill the gap about movement ecology of the species in a local/regional level.

Moreover, you report important data about individuals re-sighted. I am familiar with the oceanic manta ray population in Isla de la Plata, Ecuador, which was discovered as the biggest in the world, but most importantly, it has been noticed that the re-sighting percentage over more than 10 years is very low (M. Guerrero, unpublished data). Based on your data, could provide any discussion about this difference?

Validity of the findings

I consider the aims of the paper are clearly visualized as achieved in the conclusions. I consider that photo identification based on Marshall et al. 2009 is the standard and accurate methodology to properly identify these species and also found suitable the use of innovative technology such as drones to support their traditional surveys. Additionally, I found correct that the authors recognize that more monitoring is needed to support this information, mainly in the Yasawa Island Group so that useful data can be generated for local/regional conservation management.

Additional comments

I think the paper structure conforms with PeerJ standards except for Results and Discussion presented here as one section while they are separated in the authors guideline. This will need confirmation from the editor.

Annotated reviews are not available for download in order to protect the identity of reviewers who chose to remain anonymous.

Reviewer 2 ·

Basic reporting

This work was clear and well written. I see no major flaws within the work or the analysis.

Experimental design

observational so there are no major problems.

Validity of the findings

A solid experimental responding, no problems

---

## Round 0.2 · Minor Revisions

This new version of the manuscript has been substantially improved, relative to the first version. The new version only needs minor, but necessary, changes to be accepted. They are included in the attached PDF.

My main request is to delete Figure 5 because it is confusing and was not required by any of the reviewers in the first round. Based on the information shown in the new Figure, it became clear that the definition of "resighting" used by the authors may be problematic. Please provide the definition in the methodology section, keeping in mind that sightings of the same individual on a single day should not be considered "resightings" because they are non-independent from each other.

---

## Round 0.3 · accepted · Accept

Thanks for preparing this new version of the manuscript. In my opinion, it is ready to be published.